**Data Availability Statement:** All relevant data are within the manuscript and its Supporting Information files.

# Translation, cross-cultural adaptation and validation of Patient Satisfaction with Pharmacist Services Questionnaire (PSPSQ 2.0) into the Nepalese version in a community settings

Sunil Shrestha[1,2☯*], Binaya Sapkota[3☯*], Santosh Thapa[4‡], Bhuvan K. C.[5‡*], Saval Khanal[6‡]

1 Department of Pharmaceutical and Health Service Research, Nepal Health Research and Innovation Foundation, Lalitpur, Province Bagmati, Nepal, 2 Department of Pharmacy, Nepal Cancer Hospital and Research Center Pvt. Ltd, Harisiddhi, Lalitpur, Province Bagmati, Nepal, 3 Department of Pharmaceutical Sciences, Nobel College, Affiliated to Pokhara University, Kathmandu, Province Bagmati, Nepal, 4 Jeevee Health Pvt Ltd, Kathmandu, Province Bagmati, Nepal, 5 School of Pharmacy, Monash University Malaysia, Subang Jaya, Selangor, Malaysia, 6 Warwick Business School, University of Warwick, Coventry, United Kingdom

☯ These authors contributed equally to this work.
‡ These authors also contributed equally to this work.
* sunilcresta@gmail.com (SS); sapkota.binaya@gmail.com (BS); Bhuvan.kc@monash.edu (BKC)

## Abstract

### Background

Understanding patient satisfaction with pharmacy services can help to enhance the quality and monitoring of pharmacy services. Patient Satisfaction with Pharmacist Services Questionnaire 2.0 (PSPSQ 2.0) is a valid and reliable instrument for measuring patient satisfaction with services from the pharmacist. The availability of the PSPSQ 2.0 in Nepalese version would facilitate patient satisfaction and enhance pharmacy services in Nepal. This study aims to translate the PSPSQ 2.0 into the Nepalese version, culturally adapt it and verify its reliability and validity in the Nepalese population.

### Methods

The methodological and cross-sectional study design was used to translate, culturally adapt it, and validate PSPSQ 2.0 in Nepalese. The Nepalese version of PSPSQ 2.0 went through the full linguistic validation process and was evaluated in 300 patients visiting different community pharmacies in Kathmandu district, Nepal. Exploratory factor analysis was carried out using principal component analysis with varimax rotation, and Cronbach's alpha was used to evaluate the reliability.

**Funding:** The author(s) received no funding. Sunil Shrestha affiliated with Nepal Cancer Hospital and Research Center Pvt. Ltd provided support in the form for salary. Santosh Thapa (ST), affiliated with Jeevee Health Pvt. Ltd provided support in the form of salary. However, both organizations has no involvement in any form in this research project. Both authors have worked on this project in his capacity as an independent scholar. The authors alone are responsible for the views expressed in this article, and they do not necessarily represent the views, decisions or policies of the institution with which they are affiliated.

**Competing interests:** Nepal Cancer Hospital and Research Center Pvt. Ltd provided support in the form for salary for author Sunil Shrestha. Jeevee Health Pvt. Ltd provided support in the form of salary for author Santosh Thapa. There are no patents, products in development or marketed products associated with this research to declare. This does not alter our adherence to PLOS ONE policies on sharing data and materials.

**Abbreviations:** FDA, Food and Drug Administration; ISPOR, International Society for Pharmacoeconomics and Outcomes Research; PRO, Patient-Reported Outcomes; PSPSQ 2.0, Patient Satisfaction with Pharmacist Services Questionnaire 2.0; WHO, World Health Organization.

## Results

Three-hundred patients were recruited in this study. Participants ranged in age from 21 to 83 years; mean age was 53.93 years (SD: 15.21). 62% were females, and 34% educational level was above 12 and university level. Only 7% of the participants were illiterate. Kaiser-Meyer-Olkinwas found to be 0.696, and Bartlett's test of sphericity was significant with a chi-square test value of 3695.415. A principal axis factor analysis conducted on the 20 items with orthogonal rotation (varimax). PSPSQ 2.0 Nepalese version (20 items) had a good internal consistency (Cronbach's alpha = 0.758). Item-total correlations were reviewed for the items in each of the three domains of PSPSQ 2.0.

## Conclusion

The PSPSQ 2.0 Nepalese version demonstrated acceptable validity and reliability, which can be used in the Nepalese population for evaluating the satisfaction of patients with pharmacist services in both community pharmacy and research.

## Introduction

Patient satisfaction is an essential and widely used indicator for measuring the quality of healthcare services. The patients' judgment on whether the service provided to them meet their need and expectations are usually collected when measuring patient satisfaction [1, 2]. The data obtained from the patient satisfaction survey helps in identifying the different factors which can be used to improve and implement quality healthcare services for the patient's comfort. It is also an essential measure that informs the healthcare providers or other relevant stakeholders as a predictor of health-related behaviours expected from the patients following any health interventions [3–6].

Different tools have been developed in the past to evaluate patient satisfaction of the services delivered at community pharmacies. In 1983, the Patient Satisfaction Questionnaire (PSQ) was developed to assess patient satisfaction with medical care [7]. Gourley et al. developed the Pharmaceutical Care Satisfaction Questionnaire (PCSQ) to measure consumer satisfaction with pharmacy services [8]. In 2015, Sakharkar et al. developed an instrument called "Patient Satisfaction with Pharmacist Services Questionnaire" (PSPSQ 2.0) for measuring satisfaction of patient with pharmacist delivered services that evaluated the quality of the services. This tool helped to improve the healthcare of patients and promote patient wellness in chronic diseases [9].

Pharmacy services have evolved from a state where the role of pharmacists was narrowly focused on dispensing medicines to an age where pharmacy services cover a wide range of pharmaceutical care services with a focus on patient-centred care [10, 11]. Although, the pharmacy services have advanced significantly in high-income countries; however, the same cannot be said in case of low-and middle- income countries like Nepal [12]. Pharmacy services in Nepal are still limited to the dispensing of medicines and counselling to a large extent. According to the Department of Drug Administration (DDA), at present (February 2020), around 14,000 pharmacies (community pharmacies) are registered with the DDA [13]. Despite the dismal state of pharmacy services, there is an increase in the number of pharmacy workforce in Nepal as a result of an increase in the number of pharmacy colleges, mainly in the private sector. This rise in the number of pharmacy workforce has led to more pharmacists and

assistant pharmacists joining community pharmacies. So, the services provided in a community pharmacy sere will be an interesting area to explore and develop further. In recent years, some community pharmacies have already started providing services such as medication and lifestyle counselling, and management and reporting of adverse drug reactions (ADRs) [14–16]. This recent development in community pharmacy practice highlights the need for some standard instruments to measure the quality of services provided by community pharmacists. The majority of the people in Nepal speak and understand the Nepalese language so, translated and validated Nepaleseinstrument for measuring patient satisfaction about the pharmacy services can have a huge scope in research and also in quality improvement programs within those pharmacies.

Generally, many tools (questionnaires) are available in English. There is a common practice of translation, cultural adaption and validation of the questionnaire/instrument tools to a different language to suit the local context. There are some tools like the World Health Organization-Health-related quality of life, European Organization for Research and Treatment of Cancer, Functional Assessment of Chronic Illness Therapy, etc. which have been translated into many languages. This suggests that translating the questionnaire to the local language is very common and essential. The PSPSQ 2.0 instrument, developed by Sakharkar et al., has subsequently been translated and cross-culturally adapted in Malaysian language with acceptable measurement properties [17]. However, it has not been translated into Nepalese, and presently, there is no specific tool to measure patient satisfaction with pharmacist services available in the Nepalese language.

Nepal is a landlocked country situated between India and China, and the Nepalese language (also called the Nepali language) is an official and most commonly spoken language of Nepal. Nepalese language is spoken as a mother tongue by almost half of the total population (approximately 29 million) in Nepal [18, 19]. Around 5–7 million Nepalese language speakers are estimated to be living in India, and Nepalese language is one of the 22 scheduled languages of India [20]. Similarly, more than one-third of the whole Bhutanese population and few parts of Myanmar can speak Nepalese language [25]. Nepalese language is also the mother tongue of Bhutanese refugees living in Nepal. Nepal is a homeland to the people with unique culture, language, health literacy, socio-economic profile and health-seeking behaviour; and health practice in Nepal is very different from the English-speaking countries. For example, patients asking pharmacists for specific medicines may be a common thing in developed countries, as the population health literacy in those countries is higher. Contrary to that, pharmacists in developing countries like Nepal often receive requests from the patients for a medicine with a particular colour, brand and dosage form due to inadequate health literacy. Health literacy in the Nepalese people is one of the less explored areas [21].

To best of our knowledge, no evidence exists concerning patient satisfaction with pharmacy services in Nepal. Moreover, as explained earlier, there is an unavailability of the comprehensive, reliable and valid instrument for assessing patient satisfaction with pharmacy services in Nepalese. This study aimed to translate the Patient Satisfaction with Pharmacist Services Questionnaire (PSPSQ 2.0) into the Nepalese version, culturally adapt it and verify its reliability and validity in the community setting.

## Methods

### Study design and settings

This study was a methodological and cross-sectional study designed to translate the tool, culturally adapt it and verify its reliability and validity to assess the patient satisfaction with

pharmacy services (PSPSQ 2.0), using the tool developed by Sakharkar et al. [9], into a Nepalese version.

## Study site and study duration

The study site included three different community pharmacies of Kathmandu city, Nepal. Kathmandu is a capital city, situated in Bagmati Province of Nepal. The services provided by these community pharmacies are only limited to dispensing and medication counselling. The data collection took place between April 2019 and October 2019. Pharmacists and assistant pharmacists working on those pharmacies were responsible for dispensing over-the-counter, and prescription medications and they were also providing medication counselling. They were also responsible for stock and inventory management. The community pharmacies in the study were not associated with any specialised clinics or general clinics such as diabetes and psychiatric clinics.

## Study population, inclusion and exclusion criteria

Patients receiving pharmacy services from different community pharmacies were included in this study. Patients were included in this study if they met the following inclusion criteria: (a) native Nepalese, (b) able to understand Nepalese, (c) patients receiving pharmacy-related services for at least three months from the same pharmacist and community pharmacy, (d) aged over 18 years and (e) does not have a psychiatric illness. Patients receiving services from the pharmacists working at the hospital and clinical settings were excluded in this study.

## Instruments

**Original PSPSQ 2.0.**   PSPSQ 2.0 [9] developed by Shahakar et al., has been evaluated psychometrically. It is a validated and reliable instrument for measuring patient satisfaction with pharmacist services (see S1 File). This tool consists of 20 items and divided into three domains, i.e. quality of care, pharmacist-patient relationship and overall satisfaction using a four-point, Likert-type scale [9]. The first domain 'quality of care' comprised of 10 items. The second domain is the pharmacist-patient relationship which contains six items. The final domain is overall satisfaction which comprised of 4 items.

**Demographic questionnaire.**   The questionnaire was developed, comprising of six items questionnaire that explored demographic and related information of patients: age, gender, educational level, working status, ethnicity and religion.

**Translation, cultural adaptation and validation.**   The methods for translation, cultural adaptation, validation and reliability are described briefly under respective subheadings after a paragraph on PSPSQ (2.0).

## Step 1—Translation procedures and cultural adaptation

The PSPSQ 2.0 was used as a study tool to translate it into the Nepalese version. Before translation, the formal permission to translate, culturally adapt it, verify its reliability and validity of the instrument PSPSQ 2.0 was taken from authors of Sakharkar et al. (2015) via email. The process of translation and cultural adaptation of PSPSQ 2.0 questionnaire was conducted according to standard translation guidelines, i.e. FDA PRO Guidance [22] and ISPOR Good Practice Guidelines for linguistic and cultural adaptation and validation [23] as prescribed. The research team did the translation with the help of accredited translators. The translation and cross-cultural adaptation process was done in five different stages–

1. At the first stage of the translation (forward translation) process, the PSPSQ 2.0 English version was sent to two independent bilingual translators who were born in Nepal and native Nepalese speakers to translate English to Nepalese.

2. The second stage is reconciliation stage where the two Nepalese forward translations by two different, forward translators were reconciled by research and translation coordinator (SS) along with study team member (ST), to deliver a single "reconciled version" of the translated questionnaire.

3. Third, the reconciled version was then sent to two independent bilingual translators blinded to the original English version and having English as their first language for the backward translations.;

4. Fourth, an expert committee was formed by the study team for cultural adaptation. The expert committee comprised of pharmacy academician, pharmacists from different settings, i.e. community, clinical and hospital, forward and backward translators). Experts were contacted by the research coordinator on personal approach within the country and after their consent committee was formed. This committee then compared the backward translations with the original English version and made relevant changes, improvements and cross-cultural adaptations in order to produce the version that was used in the pilot study. After reaching consensus, the committee approved an "intermediate version" of the translated Nepalese version;

5. In the last step, the version produced in the fourth stage was subjected to a pre-test to ensure proper comprehension of each question and cultural appropriateness testing of the intermediate version was done by the pre-testing and concluded with the "final version", the final Nepalese version of PSPSQ 2.0.

## Step 2—Validation and reliability analysis

**Face validity.** The face validation of the PSPSQ 2.0 was performed by collecting feedback from the participants of the pilot study (n = 15). During the pilot study, the intermediate version was administered to the fifteen patients comprised of native Nepalese speakers. Data collection form along with the demographic questionnaire and Nepalese version PSPSQ 2.0 developed was used to collect qualitative information following an interview with the pilot participants. Participants responded to the PSPSQ 2.0 Nepalese version and then evaluated it for intelligibility, appearance, clarity, and wording. Participants were asked if they encountered any difficulty and/or confusion on understanding the questions. They were also asked about any problematic or upsetting words which they might have found in the questionnaire. Participants were also encouraged to give suggestions for improvements and allowed to present the question in an alternative way where they considered it appropriate.

**Content validity.** This tool, in its original language, was already evaluated for content validity [9]. This questionnaire was already used by various experts to assess the satisfaction of the people on the pharmacists-delivered services. Hence, we assumed the tool is already validated for its content, and the questionnaire already contains items from the desired content domains. Therefore, we did not perform any content validity tests on our own.

**Data collection process.** A self-administered survey was conducted for evaluating the construct validity and the internal consistency of the tool. Patients visiting the community pharmacies and receiving the pharmacy services from the same pharmacies were approached,

and the aim of the study was explained. Written informed consent from the participants was taken and was assured that their participation in this study was voluntary, and confidentiality of them will be maintained. Data collection form along with the demographic questionnaire and the Nepalese PSPSQ 2.0 was given to the patients receiving pharmacy services from different community pharmacies. The data collection form was distributed by one investigator of this study in each community pharmacy, and patients were given 15–30 minutes to fill with the questionnaire by themselves. The questionnaire was only given after patients left the community pharmacy. The investigator of the study assisted those patients who could not read and write by explaining when needed while filling up the questionnaire. Additional time was given to participants to fill with the questionnaire when required.

**Sample size calculation for the survey.** Using the item-response theory (IRT), the sample size required for the study was calculated [24]. Several studies have suggested different sample sizes. Some studies have recommended an item-to-respondent ratio of 1:5 up to 1:10, respectively [25, 26]. However, we have gathered data from 300 patients as PSPSQ 2.0 consisted of 20 items, thereby refining the ratio to 1:15. Three-hundred samples were sufficient for this study and were enough to produce reasonable factor solutions. However, at least 120 participants are required in conducting a factor analysis [27].

**Sampling adequacy and sphericity.** Before the performance of exploratory for its appropriateness in the factor analysis, the sampling adequacy was analysed using the Kaiser-Meyer-Olkin (KMO). KMO has to be more than 0.5 to be considered acceptable. Bartlett's test of sphericity was completed to figure out the common factors and to specify the appropriateness of the factor analysis model [28, 29].

**Construct validity.** The construct validity of the tool was checked with exploratory factor analysis (EFA) with varimax rotation on the 20 items of the questionnaire. It was intended to evaluate whether correlations among items were >0.3 and also to check the factorability of the correlation matrix using Bartlett's test of sphericity [30]. EFA with Kaiser-Meyer-Olkin (KMO) measure of sampling adequacy and Bartlett's test of sphericity supported the validity of the PSPSQ 2.0 questionnaire. Eigenvalues were taken out first. The potential number of factors was determined by the number of factors with eigenvalues greater than one [30, 31] and by the visual inspection of the scree plot with the inflection in the slope. Eigenvalues associated with each factor before and after extraction, and after rotation were also analysed. The eigenvalues associated with each factor represented the variance explained by that particular factor: the eigenvalue was translated into the percentage of variance explained.

**Reliability and internal consistency measurement.** Cronbach's alpha coefficient was used to measure the internal consistency and reliability of the dimensions of the questionnaire. A coefficient value greater than 0.70 indicates a high level of internal consistency. Alpha values ≥0.7 were considered satisfactory [32].

**Other statistical analysis.** Data were entered, cleaned and analysed using the Statistical Package for Social Sciences (IBM SPSS Statistics, Armonk, NY, IBM Corp) Version 21.0.

## Ethical consideration

The ethics approval was taken obtained from the Institutional Review Committee (IRC) of Nobel College, Kathmandu, Nepal (Reference number was EPY IRC 160/2018). Written consent was taken from the participants of the study and was assured that their participation in this study was voluntary, and confidentiality will be maintained. Consent from the participating experts was also taken to form an expert review committee in written form.

# Results

## Patients' socio-demographic characteristics

The socio-demographic characteristics of the patients are shown in Table 1. Participants ranged in age from 21 years to 83 years; the mean age was 53.93 years (*SD*: 15.21). Majority of the participants(62%) were females. Almost one-third of participants' educational level was above 12 and the university level, which was followed by primary level (22.7) and secondary level (22.3%). Only 7% of the participants were found to be illiterate. Regarding the working status of participants, the majority (81.7%) of participants' status was working.

## Translation and cultural adaptation

The process of translation and cultural adaptation generated the Nepalese version of PSPSQ 2.0 (See S2 File). During the process, no significant difficulties were found. However, some negligible changes in grammatical structures were needed. The pilot testing showed that there was no any difficulty among participants regarding the understanding of all 20 items of the PSPSQ 2.0. The expert review committee performed a cultural adaptation.

## R-matrix (i.e., the correlation matrix)

R-matrix (i.e., the correlation matrix)—the top half contained the correlation coefficient between all pairs of questions, and the bottom half one-tailed *p*-values of them (Shown in S1 Table). For the straightforward interpretation, only the columns for the first and last five questions in the questionnaire were displayed. Variables with very few correlations above 0.3 might

**Table 1. Socio-demographic characteristics of patients.**

| Study characteristics | N | Percent (%) |
|---|---|---|
| **Age** | Mean (SD): 53.93 (15.21) | |
| | Minimum- Maximum: 21–83 | |
| **Gender** | | |
| Male | 114 | 38.0 |
| Female | 186 | 62.0 |
| **Education Level** | | |
| Illiterate | 21 | 7.0 |
| Informal education | 42 | 14.0 |
| Primary level (up to 5) | 68 | 22.7 |
| Secondary level (6–12) | 67 | 22.3 |
| Above 12 and University level | 102 | 34.0 |
| **Working Status** | | |
| Working | 245 | 81.7 |
| Not Working | 55 | 18.3 |
| **Ethnicity** | | |
| Brahmin / Chettri | 141 | 47.0 |
| Janjati | 140 | 46.7 |
| Others | 19 | 6.3 |
| **Religion** | | |
| Hindu | 209 | 69.7 |
| Buddhist | 47 | 15.7 |
| Christian | 28 | 9.3 |
| Muslim | 16 | 5.3 |

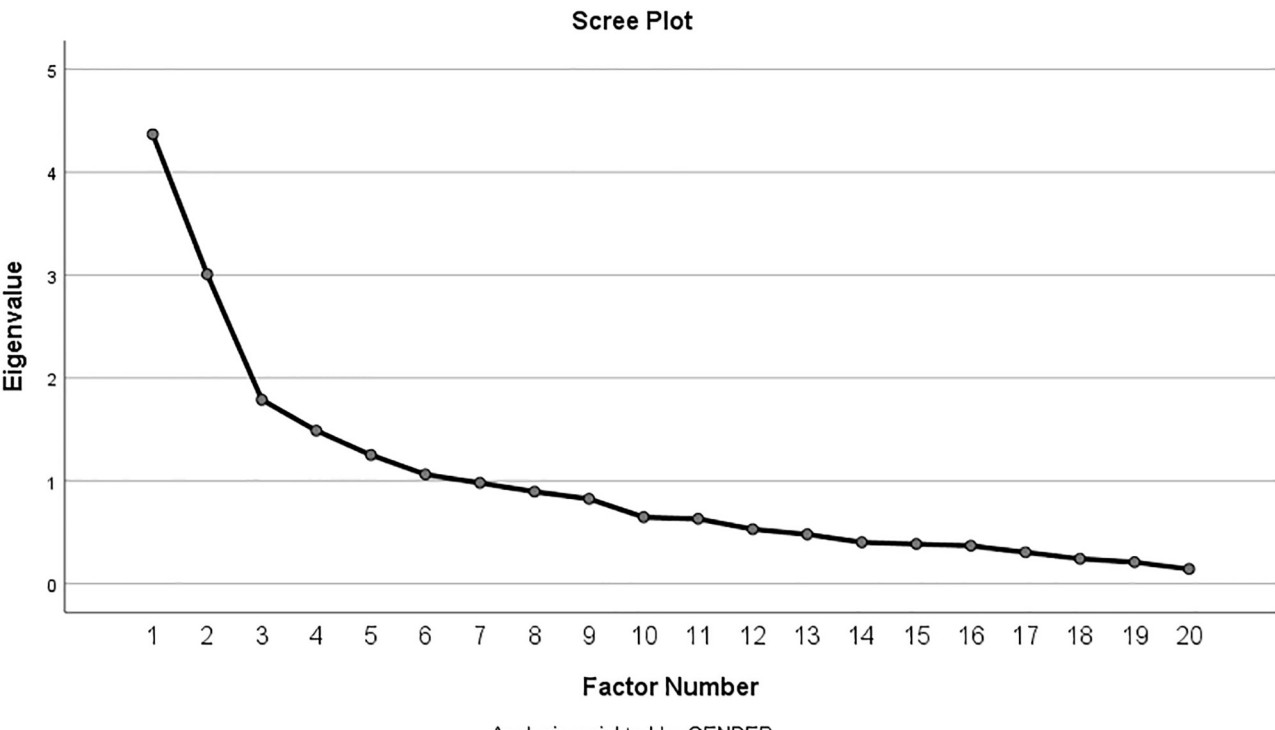

**Fig 1. Scree plot.**

not fit with the pool of items, and variables with correlations greater than 0.9 might be collinear. All questions correlated reasonably well with all; therefore, any questions from Q1 to Q20 were not eliminated at this stage.

## Sampling adequacy and sphericity

The results of the KMO statistic test was found to be 0.696. This showed that the sample size was probably adequate for factor analysis. Bartlett's test of sphericity was significant with a chi-square test value of 3695.415 (degree of freedom = 190, $p < 0.0005$) indicating that factor analysis was adequate to the observed data.

**Construct validity.** In summary, a principal axis factor analysis (FA) with orthogonal rotation (varimax) was executed on all the 20 items. Six factors had eigenvalues greater than the Kaiser's criterion of 1 and overall explained 64.80% of the variance. As the sample size was sufficient as per the calculation, and scree plot and Kaiser's criterion converged on this value, six factors were retained. (Fig 1).

## Total variance explained

Table 2 shows the eigenvalues associated with each factor before and after extraction, and after rotation. All 20 factors were identified before extraction (i.e., as many eigenvectors as the variables). Eigenvalues associated with each factor represented the variance explained by that factor. Eigenvalue was then converted into the percentage of variance explained (e.g., factor 1 explained 21.84% of total variance). The first few factors explained a large amount of variance (especially factors 1 and 2) and the subsequent factors small amount. All factors with eigenvalues >1 were extracted, which gave six factors. Rotation also ensured the factor structure

**Table 2. Total variance explained.**

| Factor | Initial Eigenvalues | | | Extraction Sums of Squared Loadings | | | Rotation Sums of Squared Loadings | | |
|---|---|---|---|---|---|---|---|---|---|
| | Total | % of Variance | Cumulative % | Total | % of Variance | Cumulative % | Total | % of Variance | Cumulative % |
| 1 | 4.368 | 21.840 | 21.840 | 3.977 | 19.884 | 19.884 | 3.198 | 15.990 | 15.990 |
| 2 | 3.007 | 15.034 | 36.874 | 2.492 | 12.461 | 32.345 | 1.743 | 8.713 | 24.703 |
| 3 | 1.787 | 8.936 | 45.810 | 1.254 | 6.270 | 38.616 | 1.742 | 8.710 | 33.412 |
| 4 | 1.487 | 7.437 | 53.247 | 1.017 | 5.083 | 43.698 | 1.256 | 6.281 | 39.693 |
| 5 | 1.251 | 6.254 | 59.501 | .688 | 3.441 | 47.140 | 1.212 | 6.060 | 45.753 |
| 6 | 1.061 | 5.307 | 64.808 | .646 | 3.230 | 50.369 | .923 | 4.616 | 50.369 |
| 7 | .980 | 4.900 | 69.707 | | | | | | |
| 8 | .894 | 4.470 | 74.177 | | | | | | |
| 9 | .824 | 4.121 | 78.298 | | | | | | |
| 10 | .646 | 3.232 | 81.530 | | | | | | |
| 11 | .630 | 3.151 | 84.681 | | | | | | |
| 12 | .530 | 2.648 | 87.328 | | | | | | |
| 13 | .479 | 2.395 | 89.724 | | | | | | |
| 14 | .402 | 2.010 | 91.734 | | | | | | |
| 15 | .385 | 1.927 | 93.660 | | | | | | |
| 16 | .369 | 1.846 | 95.506 | | | | | | |
| 17 | .305 | 1.524 | 97.031 | | | | | | |
| 18 | .242 | 1.211 | 98.241 | | | | | | |
| 19 | .209 | 1.045 | 99.286 | | | | | | |
| 20 | .143 | .714 | 100.000 | | | | | | |

Extraction Method: Principal Axis Factoring.

showing the importance of the six factors in the analysis. Factor 1 accounted for more variance (19.88%) than other five (12.46%, 6.27%, 5.08%, 3.44% and 3.23%) before rotation. After rotation it accounted for 15.99% of variance (compared to 8.71%, 8.71%, 6.28%, 6.06% and 4.61% of rest of the factors).

## Reliability and validity

Cronbach's alpha for the reliability of all the 20 items was 0.758, which indicated very good reliability while that for each of the domains ranged from 0.621 to 0.845 (Shown in Table 3). Cronbach's alpha value was found to be 0.845, 0.683 and 0.621 for the domain quality of care, patient-pharmacist relationship and overall respectively.

## Discussion

PSPSQ 2.0 was successfully translated, culturally adapted, and its reliability and validity into Nepalese version were verified using the established methodology [22, 23]. The good comprehensibility and simplicity of completion of PSPSQ 2.0 Nepalese version were reinforced by the feedback from the participants included in the pilot study and reproduce the evaluation of the original English version in this regard [9]. This Nepalese version also demonstrated acceptable psychometric properties of reliability and validity for the evaluation. This research has produced a Nepalese version of the PSPSQ 2.0 which, after transcultural adaptation and validation has proven to be a discriminant, valid and reliable tool to assess patients' satisfaction with pharmacy services.

**Table 3. Item-total reliability statistics of quality of care, interpersonal relationship (pharmacist/patient) and overall domain and reliability analysis.**

| Domain | Items | Scale Mean if Item Deleted | Scale Variance if Item Deleted | Corrected Item-Total Correlation | Squared Multiple Correlation | Cronbach's Alpha if Item Deleted | Cronbach's Alpha |
|---|---|---|---|---|---|---|---|
| Quality of Care | Q1 | 27.24 | 27.515 | .404 | .432 | .842 | **0.845** |
| | Q2 | 27.11 | 26.984 | .442 | .384 | .839 | |
| | Q3 | 27.31 | 25.220 | .578 | .547 | .828 | |
| | Q4 | 27.42 | 22.801 | .678 | .656 | .817 | |
| | Q5 | 27.53 | 22.794 | .765 | .763 | .808 | |
| | Q6 | 27.45 | 24.413 | .646 | .636 | .821 | |
| | Q7 | 27.29 | 27.204 | .395 | .376 | .843 | |
| | Q8 | 27.38 | 23.000 | .699 | .580 | .815 | |
| | Q9 | 27.34 | 27.940 | .230 | .364 | .859 | |
| | Q10 | 27.13 | 25.921 | .582 | .502 | .829 | |
| Interpersonal Relationship (pharmacist/patient) | Q11 | 15.98 | 6.532 | .219 | .190 | .714 | **0.683** |
| | Q12 | 15.80 | 5.943 | .445 | .301 | .631 | |
| | Q13 | 15.74 | 6.372 | .447 | .332 | .635 | |
| | Q14 | 15.77 | 5.985 | .437 | .341 | .634 | |
| | Q15 | 15.85 | 5.548 | .566 | .432 | .587 | |
| | Q16 | 15.76 | 6.189 | .410 | .329 | .643 | |
| Overall | Q17 | 9.07 | 3.065 | .518 | .310 | .491 | **0.621** |
| | Q18 | 9.13 | 3.223 | .476 | .386 | .522 | |
| | Q19 | 9.21 | 2.841 | .452 | .304 | .513 | |
| | Q20 | 9.87 | 2.516 | .285 | .125 | .705 | |
| **Pooled (All 20 Items)** | | | | | | | **0.758** |

The practising pharmacists working in community settings of Nepal or specific geographical regions of other countries (e.g. part of India, Bhutan, and Myanmar) where Nepalese is spoken predominantly can be significantly benefitted by this translated, culturally adapted and validated reliable questionnaire. There are people with Nepalese ethnicity in India, Bhutan and Myanmar. They belong to the broader Nepalese community but citizens of a different country. While there might be some similarity in terms of culture, language, festival etc. when it comes to healthcare seeking practice or healthcare belief the culture or influence of culture might be different given the difference in context and access to resources and their standards of living. However, further studies are needed to comment on the overall culture and its effect on healthcare-seeking practice among Nepalese communities living Bhutan and Myanmar. Due to high mobility of people between Nepal and India, the Nepalese community living in a particular region of Northern, Eastern and North-Eastern India might have different cultural experiences when compared to the Nepalese community of Bhutan and Myanmar. Again, more studies are needed on their cultural experiences and the influence of their culture on healthcare-seeking practice or healthcare belief system.

Community pharmacies stand as the first point of contact for obtaining medicines and offer various services like counselling on diseases and medicines, dressing of wounds, administering injections etc [16, 33, 34]. Easy access, flexible opening schedule, free or minimum service charge are the reasons community pharmacies are preferred than other health care service centres. Community pharmacies in Nepal thus play a significant role in catering over the counter and prescription medicines, managing minor illness and referring patients to specialised care centres [33, 35, 36].

Patient satisfaction may not relate to consumer satisfaction, as we see it in the retail sector. In Nepal, community pharmacy is also known as a retail pharmacy. There may be two

possibilities. The first possibility is that an unsatisfied consumer may not buy a product or service; it may be detrimental to business but not to patient's health. The second possibility maybe retail sectors other than community pharmacy do different things to satisfy all consumers' needs; however, the same may not be the case for healthcare or community pharmacies. For community pharmacies, an unsatisfied patient means he or she might not have enough information, may not like the pharmacy environment as there is no provision for privacy, in that case, the patient may not be able to express his concerns or his healthcare needs, and it may affect his health outcome.

Contrary to that, a pharmacist cannot dispense whatever a patient request over the counter to make him/her happy. Patient satisfaction is a unique issue when it comes to community pharmacy services; it is an important issue. However, it needs to be defined and measured. Assessing satisfaction of pharmacy is considered as a critical indicator of the quality of pharmacy services. It will also contribute to the indirect monitoring the patient prognosis, as it reflects whether the service provided by the pharmacist is meeting patients' expectations or values [37]. This is also useful for setting a standard when launching new pharmacist-delivered services or strategies [38]. This tool will ensure the availability of one of the essential evaluation tools to the researchers and policy stakeholders performing pharmaceutical health service research or implementing any such programs to improve the patient outcomes as a result of pharmacists delivered services. In the future, this tool can be integrated with services such as medication counselling, management of adverse drug reactions, pharmaceutical care services by a pharmacist. This research fulfils the need for the comprehensive, reliable and valid instrument for assessing patient satisfaction with pharmacy services in the Nepalese version.

The results of the confirmatory factor analysis supported the validity of the PSPSQ 2.0 questionnaire. Bartlett's test of sphericity was significant ($p < 0.0001$) with a $\chi^2$ value of 3695.415. This showed that the sample size was probably acceptable for factor analysis, which was supported by the recommendations of a minimum of 100 to 200 participants by several studies [25–27].

Based on the results of the reliability analysis, the internal consistency coefficient (Cronbach alpha) for the instrument of the PSPSQ 2.0 and its three domains was found to be good, demonstrating that it can generate reliable scores. Cronbach's alpha value exceeded the pre-set value (0.70) and illustrated excellent reliability within the constructs [32]. The findings of this study were similar to the study conducted by Hassali et al., where this tool was translated into Malaysian language [17], which showed Cronbach's alpha value as 0.907, 0.762 and 0.913 for the domain "Quality of Care", 0.762 for the "patient-pharmacist relationship" and the pooled 16 items respectively. Comparing to the study by Hassali et al. [17] and our study, variability was found in Cronbach's alpha value in every domain. There were some discrepancies noted between the original English version and the Nepalese version. Cronbach's alpha was found below than the original instrument where the developer of the tool reported. Cronbach's alphas value of 0.95 in Veterans Affairs (VA) based clinics, 0.98 and 0.96 in two community-based clinics diabetes clinic and psychiatric clinic, respectively [9]. The eigenvalues and scree plot supported a 3-factorial nature of the translated questionnaire. Another discrepancy found in the study was a study site where the original study conducted at VA based clinics and community-based clinics. However, we collected data from the community pharmacies with no specialised or general clinics. A significant strength of this study is that it provided the first measure to assess the patient's satisfaction with pharmacy services in Nepal. Elaborated translations and cultural adaption procedures were conducted. Additionally, the sample size in this study is more extensive than many similar studies conducted in other countries.

This study has some limitations, as well. First, as this study was conducted among patients taking services from the community pharmacy, it could not include patient's receiving services

in more diverse environments such as clinical pharmacy and hospital pharmacy. More studies in different settings, such as clinical and hospital pharmacy, is advisable. The second limitation, the patients in this study do not represent the target population, as patients were selected based on available extraction. This would be a limitation to generalising the results of this study. The time duration taken by the patients in this survey was about 15–30 minutes to complete the Nepalese version PSPSQ 2.0 questionnaire, which is acceptable as suggested by other studies. However, some of them needed assistance in completing the questionnaire, as they were illiterate and did not know how to read and write, thus one of the drawbacks of the tool. We have included patients receiving pharmacy services from the same pharmacy for at least three months, and due to the selection criteria for the patients, it was inevitable to avoid recall bias in the study.

Nonetheless, the validity and reliability of the Nepalese version of PSPSQ 2.0 were verified as adequate in this study. Thus, the findings of this study could help to assess patient satisfaction with pharmacy services in the future and be actively used in related studies. In future, this tool can be used in community settings to study and assess the patient's satisfaction with pharmacy services. Further revisions of this instrument can be done after taking into consideration the nature of pharmacy services provided by hospital pharmacies or clinical pharmacy.

## Conclusion

The findings of the study found that the Nepalese version of PSPSQ 2.0, developed initially by Sakharkar et al.,. is a valid and reliable and useful tool to assess patient's satisfaction with pharmacy services. Accordingly, the Nepalese version of PSPSQ 2.0 could be used in the future to measure patient satisfaction with various services provided by community pharmacists among the Nepalese speaking population.

## Supporting information

**S1 File. PSPSQ 2.0 original version.**
(DOC)

**S2 File. PSPSQ 2.0 Nepalese version.**
(PDF)

**S3 File.**
(XLSX)

**S1 Table. Correlation matrix.**
(DOCX)

## Acknowledgments

We would like to thank Dr Prashant Sakharkar, Department of Clinical and Administrative Sciences, Roosevelt University, College of Pharmacy, USA and Dr Anandi V. Law Department of Pharmacy Practice and Administration, Western University of Health Sciences, College of Pharmacy, Pomona, USA for providing us with a questionnaire for a translation purpose. We would like to thank all the patients who participated during the study and suggested a few terms for the improvement of a Nepalese version of PSPSQ 2.0. We would like to thank Ms Krisha Danekhu, Nepal Health Research and Innovation Foundation, for her assistance during data collection.

## Author Contributions

**Conceptualization:** Sunil Shrestha, Santosh Thapa, Saval Khanal.

**Data curation:** Sunil Shrestha, Binaya Sapkota, Santosh Thapa.

**Formal analysis:** Binaya Sapkota.

**Funding acquisition:** Santosh Thapa.

**Investigation:** Sunil Shrestha, Saval Khanal.

**Methodology:** Sunil Shrestha, Santosh Thapa, Bhuvan K. C., Saval Khanal.

**Project administration:** Sunil Shrestha.

**Resources:** Santosh Thapa, Saval Khanal.

**Software:** Binaya Sapkota.

**Supervision:** Sunil Shrestha, Bhuvan K. C., Saval Khanal.

**Validation:** Sunil Shrestha, Binaya Sapkota, Santosh Thapa, Bhuvan K. C., Saval Khanal.

**Visualization:** Sunil Shrestha, Binaya Sapkota, Saval Khanal.

**Writing – original draft:** Sunil Shrestha, Saval Khanal.

**Writing – review & editing:** Sunil Shrestha, Binaya Sapkota, Santosh Thapa, Bhuvan K. C., Saval Khanal.

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
