## [Decision Letter · Decision Letter 0]

30 Jun 2020

PONE-D-20-14361

Translation, cross-cultural adaptation and validation of Patient Satisfaction with Pharmacist Services Questionnaire (PSPSQ 2.0) into the Nepalese version

PLOS ONE

Dear Dr. Shrestha,

Thank you for submitting your manuscript to PLOS ONE. After careful consideration, we feel that it has merit but does not fully meet PLOS ONE’s publication criteria as it currently stands. Therefore, we invite you to submit a revised version of the manuscript that addresses the points raised during the review process.

It is noted that there are several comments in common between the two reviewers. These comments need to be addressed. Additionally, please pay careful attention to the justification of six factors in light of the scree plot and original scale. Reviewer 1 suggests additional analysis would be required to support the chosen factor model.

We look forward to receiving your revised manuscript.

Kind regards,

Carl Richard Schneider, BN, BPharm (Hon), PhD

Academic Editor

PLOS ONE

Journal Requirements:

2. Please provide additional details regarding the consent of the expert reviewers employed to test the validation of this questionnaire. In the ethics statement in the Methods and online submission information, please ensure that you have specified (1) whether consent was informed and (2) what type you obtained (for instance, written or verbal, and if verbal, how it was documented and witnessed). If the need for consent was waived by the ethics committee, please include this information.

3. Please ensure that you have specified whether participant consent was informed. In addition, please refrain from stating p values as .000, either report the exact value or employ the format p<0.001.

4. We noticed you have some minor occurrence of overlapping text with the following previous publication(s), which needs to be addressed:

https://fulltxt.org/article/957

https://mhealth.jmir.org/2018/1/e24/

In your revision ensure you cite all your sources (including your own works), and quote or rephrase any duplicated text outside the methods section. Further consideration is dependent on these concerns being addressed.

"The author(s) declare that they have no competing interests."

We note that one or more of the authors are employed by a commercial company: Jeevee Health Pvt Ltd.

5.1. Please provide an amended Funding Statement declaring this commercial affiliation, as well as a statement regarding the Role of Funders in your study. If the funding organization did not play a role in the study design, data collection and analysis, decision to publish, or preparation of the manuscript and only provided financial support in the form of authors' salaries and/or research materials, please review your statements relating to the author contributions, and ensure you have specifically and accurately indicated the role(s) that these authors had in your study. You can update author roles in the Author Contributions section of the online submission form.

5.2. Please also provide an updated Competing Interests Statement declaring this commercial affiliation along with any other relevant declarations relating to employment, consultancy, patents, products in development, or marketed products, etc. 

6. We note that you have indicated that data from this study are available upon request. PLOS only allows data to be available upon request if there are legal or ethical restrictions on sharing data publicly. For information on unacceptable data access restrictions, please see http://journals.plos.org/plosone/s/data-availability#loc-unacceptable-data-access-restrictions.

Reviewers' comments:

Reviewer's Responses to Questions

**Comments to the Author**

1. Is the manuscript technically sound, and do the data support the conclusions?

Reviewer #1: Partly

Reviewer #2: Yes

2. Has the statistical analysis been performed appropriately and rigorously? 

Reviewer #1: No

Reviewer #2: Yes

3. Have the authors made all data underlying the findings in their manuscript fully available?

Reviewer #1: No

Reviewer #2: Yes

4. Is the manuscript presented in an intelligible fashion and written in standard English?

Reviewer #1: No

Reviewer #2: No

5. Review Comments to the Author

Reviewer #1: In the scarcity of the instrument of focus, the authors' aim to provide the need of validated and reliable questionnaire to measure patient's satisfaction towards pharmacy services is noble and this present study could be the answer.

However, the authors do not present the study satisfactorily and there are much room for improvement. Firstly, it seems that the manuscript was not adequately proofread before submission. The presence of basic grammatical errors in many parts of the article can be the evidence. For example, some sentences miss the full stop (for example, lines 154 and 242) and even a "sentence" is not a sentence, but just a phrase (for example, lines 146-147). The subheadings are not written consistently as well as some journal or publisher name(s) in the Reference section. The use of however and although in a sentence could be possible, but the authors seem to not use them properly and potentially create confusion for the readers (lines 96-97). Two sentences in the Introduction section are sequentially repeated in the Methods section, i.e. lines 90-93 vs 159-163. The reference number of ethical clearence presented in line 247 is slightly different from line 421.

Secondly, it seems that the authors want to provide a validated and reliable translated questionnaire that could be useful for all pharmacy settings, i.e. hospitals, clinics, and community pharmacies, but the sample was only taken from community pharmacies without sufficient details about the number of pharmacies, sampling method, and typical services provided by the pharmacies. The study may also be prone to recall bias as the defined recall period is 3 months. Providing more details on the data collection and giving more focus on community pharmacy settings (i.e specify the pharmacy setting in the title) may improve the quality of the manuscript.

Thirdly, in the Results section, there are many technical details to read the tables that usually are not supposed to be written in the paragraph. They should be put as a table caption.

Fourthly, the authors tend to repetitively report the KMO value and the adequate number of sample, especially in the Results and Disccusions sections while it may not be necessary.

Fifthly, in the construct validity analysis, the authors retained six factors in the final model, but the internal consistency reliability was calculated based on the three-factor solution like the original construct. No name was given for each of six factors retained in the model. No further discussion was given on the discrepancy of the number of domains found in the Nepalese version vs the original version. The authors also miss to report one important table, which is the distribution of each question to each factor with its corresponding factor loading. The questions should be displayed in its original language and the target language, while the authors only provide the Nepalese version in the Supplementary file.

Lastly, the authors also used visual inspection of the scree plot as it complemented the Kaiser’s criterion of 1 (K-1) to decide the number of factors retained. While they decided to use a six-factor solution, the inspection of scree plot shows that the kink of the plot is on the third eigenvalue. This result should suggest only two or three factors should be retained. Any alternative analysis, such as the parallel analysis, is recommended to support the decision for retaining six factors.

Reviewer #2: Title: Please revise; it can be shortened (remove abbreviations for instance).

Please be consistent; Napalese version vs Napalese language.

Some minor english editing of the manuscript is required.

The number of tables could be reduced.

Introduction:

Patient vs consumer satisfaction; patients needs in many ways to rely on the service; while consumers can easily disregard the service. Is this an issue when it comes to needs and whether or not to measure satisfaction?

Cultural differences between English and Nepalese what would be the most prominent ones?

Aim; I don't think as it is currently written encompass cultural adaption.

Methods:

Consistent and clearly stated.

But, from the introduction I get the impression that this tool can be used in all pharmacies; but you only collect data from community pharmacies: Perhaps it is wise to exclude some of the information provided about hospital pharmacies/clinical pharmacies; or at least this needs to be discussed thoroughly.

The sample size seems appropriate; and the statistics is well described.

Results:

The correlation matrix could be presented as a supplement in its full version.

Several of the correlation is way below 0.3, is this problematic, and should it be discussed?

Table 3, unessecary; information could be given in text.

Table 4. I suggest that you instead provide the percentages of explained variance from the different loadings.

It could also be an idea to present the scree plot before table 4; this will provide the reader with information about how many different factors one should include.

Discussion:

All data are collected by one pharmacist in community pharmacies: please discuss.

The cross cultural adaption; is the culture among Nepalese speaking people in India, Buthan and Myanmar similar?

The number of participants in the study is good. 300 is acceptable. Please do not repeat it.

Why is your sample not represenative; could you please elaborate on these characteristics. Please provide information about how many of the respondents that couldn't read or write. On p. 211 you state that patients filled out the questionnaire by themselves: pleas adjust wording.

Reference list:

Please adjust/provide adequate informartion; e.g., ref no. 33.

6. PLOS authors have the option to publish the peer review history of their article (what does this mean?). If published, this will include your full peer review and any attached files.

Reviewer #1: No

Reviewer #2: **Yes: **Kjell H. Halvorsen

---

## [Author Response · Author response to Decision Letter 0]

13 Aug 2020

We are very grateful for the reviews provided by the editor and the reviewers on our manuscript. The comments are encouraging, and the reviewers appear to share our judgment that this study and its results are interesting and relevant. Please kindly see the attached file, in blue, our detailed response to comments. 

All page numbers and line numbers refer to the revised manuscript with track changes.

Sunil Shrestha and co-authors.

---

## [Decision Letter · Decision Letter 1]

28 Sep 2020

Translation, cross-cultural adaptation and validation of Patient Satisfaction with Pharmacist Services Questionnaire (PSPSQ 2.0) into the Nepalese version in a community settings

PONE-D-20-14361R1

Dear Dr. Shrestha,

We’re pleased to inform you that your manuscript has been judged scientifically suitable for publication and will be formally accepted for publication once it meets all outstanding technical requirements.

Kind regards,

Carl Richard Schneider, BN, BPharm (Hon), PhD

Academic Editor

PLOS ONE

Additional Editor Comments (optional):

Reviewers' comments:

Reviewer's Responses to Questions

**Comments to the Author**

1. If the authors have adequately addressed your comments raised in a previous round of review and you feel that this manuscript is now acceptable for publication, you may indicate that here to bypass the “Comments to the Author” section, enter your conflict of interest statement in the “Confidential to Editor” section, and submit your "Accept" recommendation.

Reviewer #1: All comments have been addressed

Reviewer #2: All comments have been addressed

2. Is the manuscript technically sound, and do the data support the conclusions?

Reviewer #1: Yes

Reviewer #2: Yes

3. Has the statistical analysis been performed appropriately and rigorously? 

Reviewer #1: Yes

Reviewer #2: Yes

4. Have the authors made all data underlying the findings in their manuscript fully available?

Reviewer #1: Yes

Reviewer #2: Yes

5. Is the manuscript presented in an intelligible fashion and written in standard English?

Reviewer #1: Yes

Reviewer #2: Yes

6. Review Comments to the Author

Reviewer #1: Thank you for responding to my comments. Hopefully, the instrument will be useful for pharmacy practice in Nepal.

Reviewer #2: I think the authors have addressed my initial concerns appropriately, although there are still a few typos and minor errors in punctuation.

7. PLOS authors have the option to publish the peer review history of their article (what does this mean?). If published, this will include your full peer review and any attached files.

Reviewer #1: **Yes: **Antonius Nugraha Widhi Pratama

Reviewer #2: No

---

## [Editor Report · Acceptance letter]

1 Oct 2020

PONE-D-20-14361R1 

Translation, cross-cultural adaptation and validation of Patient Satisfaction with Pharmacist Services Questionnaire (PSPSQ 2.0) into the Nepalese version in a community settings 

Dear Dr. Shrestha:

I'm pleased to inform you that your manuscript has been deemed suitable for publication in PLOS ONE. Congratulations! Your manuscript is now with our production department. 

Kind regards, 

on behalf of

Dr. Carl Richard Schneider 

Academic Editor

PLOS ONE